# Burden of drug-resistant tuberculosis among contacts of index cases: a protocol for a systematic review

Temesgen Yihunie Akalu [1,2,3] Archie C A Clements,[3,4]
Eyob Alemayehu Gebreyohannes [3,5] Haileab Fekadu Wolde,[1,2,3]
Fasil Wagnaw Shiferaw,[6] Kefyalew Addis Alene[2,3]

¹Epidemiology and Biostatistics, University of Gondar, Gondar, Ethiopia
²Curtin University Faculty of Health Sciences, Perth, Western Australia, Australia
³Telethon Kids Institute, Nedlands, Western Australia, Australia
⁴Peninsula Medical School, University of Plymouth, Plymouth, UK
⁵School of Allied Health, University of Western Australia, Perth, Western Australia, Australia
⁶Australian National University, Canberra, Australian Capital Territory, Australia

**Correspondence to**
Temesgen Yihunie Akalu;
temesgenyihunie@gmail.com

## ABSTRACT

**Introduction** People having close contact with drug-resistant tuberculosis (DR-TB) patients are at increased risk of contracting and developing the disease. However, no comprehensive review has been undertaken to estimate the burden of DR-TB among contacts of DR-TB patients. Therefore, the current systematic review will quantify the prevalence and incidence of DR-TB among contacts of DR-TB patients.

**Method and analysis** Systematic searches will be conducted in Medline, Embase, Web of Science, Scopus, Cochrane Central Register of Controlled trials (CENTRAL) and Cumulative Index to Nursing and Allied Health Literature (CINHAL) databases. The search will be conducted without restrictions on time, language and geography. A random-effects meta-analysis will be conducted for effect estimates. The pooled prevalence and incidence of DR-TB will be compared between people with and without contact with DR-TB patients. The presence of heterogeneity between studies will be assessed by Higgins I² statistics. Subgroup analysis will be conducted to determine the source of heterogeneity. The risk of bias will be assessed using a visual inspection of the funnel plot and Egger's regression test statistics. Trim and fill analysis will be done in the presence of publication bias. A sensitivity analysis will be conducted by trimming low-quality studies. The systematic review will be reported according to Preferred Reporting Items for Systematic Reviews and Meta-Analyses Protocol guidelines.

**Ethics and dissemination** Ethical approval will not be required for this study as it will be a systematic review and meta-analysis based on previously published evidence. The findings of the systematic review will be presented at scientific conferences and published in scientific journals.

**Protocol registration** The protocol is published in PROSPERO with registration number CRD42023390339.

## BACKGROUND

Drug-resistant tuberculosis (DR-TB) is an important public health concern. It is defined as resistance to any of the anti-TB drugs, and it can be classified into monoresistant (resistant to only one anti-TB drug), multidrug-resistant tuberculosis (MDR-TB, resistant to both isoniazid and rifampicin), poly-resistant (resistant to more than two first-line drugs

**STRENGTHS AND LIMITATIONS OF THIS STUDY**

⇒ The review will use a comprehensive search strategy to obtain unbiased summary.
⇒ Subgroup analysis will be performed to compare the prevalence and incidence of DR-TB by study characteristics.
⇒ Findings will be reported according to Preferred Reporting Items for Systematic Reviews and Meta-Analyses Protocol.
⇒ The search will be conducted without time and geographical restrictions.
⇒ Substantial heterogeneity among included studies may be the possible limitation of the study.

except combined resistance to both isoniazid and rifampin), pre-XDR-TB (MDR-TB with resistance to either a fluoroquinolone or at least one of three injectable second-line TB drugs, but not both) and extensively drug-resistant (XDR-TB, MDR-TB with resistance to any fluoroquinolone and at least one of the second-line injectable drugs).[1] In 2021, approximately half a million people were diagnosed with DR-TB, and nearly 3.9% of new TB cases and 20% of previously treated cases were DR-TB. Three countries alone carry 42% of the global DR-TB burden in 2021: India (26%), the Russian Federation (8.5%) and Pakistan (7.9%).[2]

Contact investigation is an active case detection approach among contacts of drug-susceptible TB (DS-TB) and DR-TB patients, and its primary is to foster early diagnosis and treatment. This will interrupt disease transmission, slowing down the progression of the disease; preventing long-term irreversible physical and mental health complications, as well as social, quality of life and financial harms; and reducing the overall mortality from DR-TB.[3–5] The treatment of MDR-TB is costly and toxic and takes an average treatment duration of 2 years.[6 7] Active case finding is recommended for people having

a history of exposure to DR-TB cases as they are at a higher risk of developing the disease than the general population.[8] However, the probability of developing DR-TB among contacts will vary and depends on the infectiousness of the index case,[9] duration of contact,[9] proximity to the index case[10] and susceptibility of the contact.[11] As a result, the timing of the disease occurrence among contacts varies from as short as 6 weeks to several years.[12]

High-income countries, where the incidence of DR-TB is low in the general population, have standard practices regarding DR-TB contact investigation.[13] Approaches including radiological investigation, sputum culture, drug susceptibility tests (DST) and sophisticated genomic methods (eg, targeted next-generation sequencing (tNGS)) are used in identifying DR-TB cases among contacts of DR-TB.[14 15] Tuberculin skin test (TST) and interferon-gamma tests are used in latent TB case detection.[16 17] However, DR-TB contact screening among contacts of DR-TB patients is very limited in low-income countries due to scarce resources, where the incidence of DS-TB and DR-TB is high.[18] Recently, a growing interest in contact screening practices among contacts of DR-TB patients in low-income countries has been reported.[19]

Several systematic reviews have estimated the burden of DS-TB among people who were close contacts of DS-TB cases. Those studies showed that people having close contact with DR-TB patients are at increased risk of contracting and developing the disease. For example, a previous systematic review conducted in high-income countries in 2005 by Morrison *et al* showed that the overall burden of TB (both DS-TB and DR-TB) among contacts was 4.5%. However, the study lacked a stratified analysis of high-risk groups such as DR-TB close contacts and addressed only the prevalence of TB overall.[20] Another systematic review conducted in low-income countries in 2013 by Fox *et al* among contacts of TB patients (DS-TB and DR-TB combined) showed that the overall prevalence of TB was 3.1%.[4] The findings from previous studies have provided inconclusive evidence and are now outdated.[21] Therefore, the current systematic review will quantify the burden of DR-TB among people in contact with DR-TB patients including household, close and casual contacts of DR-TB patients. The primary objective is to quantify the pooled proportion of DR-TB among people in close contact with DR-TB patients. Our secondary objective is to assess study-level characteristics that may be associated with a high proportion of DR-TB.

### Review questions
What is the prevalence of DR-TB among contacts of DR-TB patients?

What is the incidence of DR-TB among contacts of DR-TB patients?

What are the study-level characteristics associated with high prevalence and incidences of DR-TB among contacts of DR-TB patients?

**Table 1** Proposed search strategy in Medline

| Search | Query |
|---|---|
| #1 | ('multidrug-resistant* tuberculosis' or 'multidrug-resistant* TB' or 'extensively drug-resistant*' or 'drug-resistant* tuberculosis' or 'MDR-TB' or 'XDR-TB' or 'DR-TB').mp.(mp=title, book title, abstract, original title, name of substance word, subject heading word, floating sub-heading word, keyword heading word, organism supplementary concept) word, protocol supplementary concept) word, rare disease supplementary concept) word, unique identifier, synonyms) |
| #2 | ('tracing' or 'contact*' or 'investigation' or 'household' or 'screening' or 'infectious disease contact screening' or 'household contact' or 'close contact*' or 'partner notification*' or 'index case*').mp.(mp=title, book title, abstract, original title, name of substance word, subject heading word, floating sub-heading word, keyword heading word, organism supplementary concept) word, protocol supplementary concept) word, rare disease supplementary concept) word, unique identifier, synonyms) |
| #3 | 1 AND 2 |

DR-TB, Drug-resistant Tuberculosis; DS-TB, Drug-susceptible Tuberculosis; MDR-TB, Multidrug-resistant Tuberculosis; PRISMA, Preferred Reporting Items for Systematic Reviews and Meta-Analyses Protocols; TB, Tuberculosis; XDR-TB, Extensive Drug Resistant Tuberculosis.

## METHODS
### Protocol registration
The protocol for this systematic review is registered in PROSPERO with a protocol registration number CRD42023390339 and reported according to the Preferred Reporting Items for Systematic Reviews and Meta-Analyses Protocols (PRISMA-P) statement 2015.[22] The article screening and selection processes will be reported using the PRISMA-20 flow chart (online supplemental file 1).

### Search strategy
Systematic searches will be conducted in Medline (via OVID), Embase, Web of Science, Scopus and Cumulative Index to Nursing and Allied Health Literature (CINAHL) databases. We will use the Cochrane Central Register of Controlled Trials (CENTRAL) database to search for experimental and quasi-experimental studies. Other search engines such as Google and Google Scholar will be searched for grey literature. The search will be conducted from the inception of each database without restrictions on time and geography. We will also perform hand-searching of the reference lists of included studies. When additional information is required, we will contact the corresponding authors. The search strategy for Medline is summarised in table 1.

## Eligibility criteria

All studies reporting the burden (ie, proportion, prevalence or incidence) of DR-TB among people with contacts (ie, households, close and casual contacts) of DR-TB will be included in this systematic review and meta-analysis. We will exclude reviews, commentaries, editorials, case reports and case series and animal studies. Moreover, studies that lack information on the outcome variable and are conducted only on DS-TB patients will be excluded. Studies will be included based on the PICO (population, intervention, comparator and outcome) framework.

## Outcome measures
### Primary outcome measures

The primary outcomes of the study are the prevalence and incidence of DR-TB among people having contact with DR-TB patients. The incidence of DR-TB among people having contact with DR-TB patients will be calculated by the year of enrolment. The prevalence or incidence of DR-TB among people having contact with DR-TB will be determined. Contact will be defined as a person living in the same household as the index case or exposure to DR-TB patients in transportation, workplace and recreational sites.

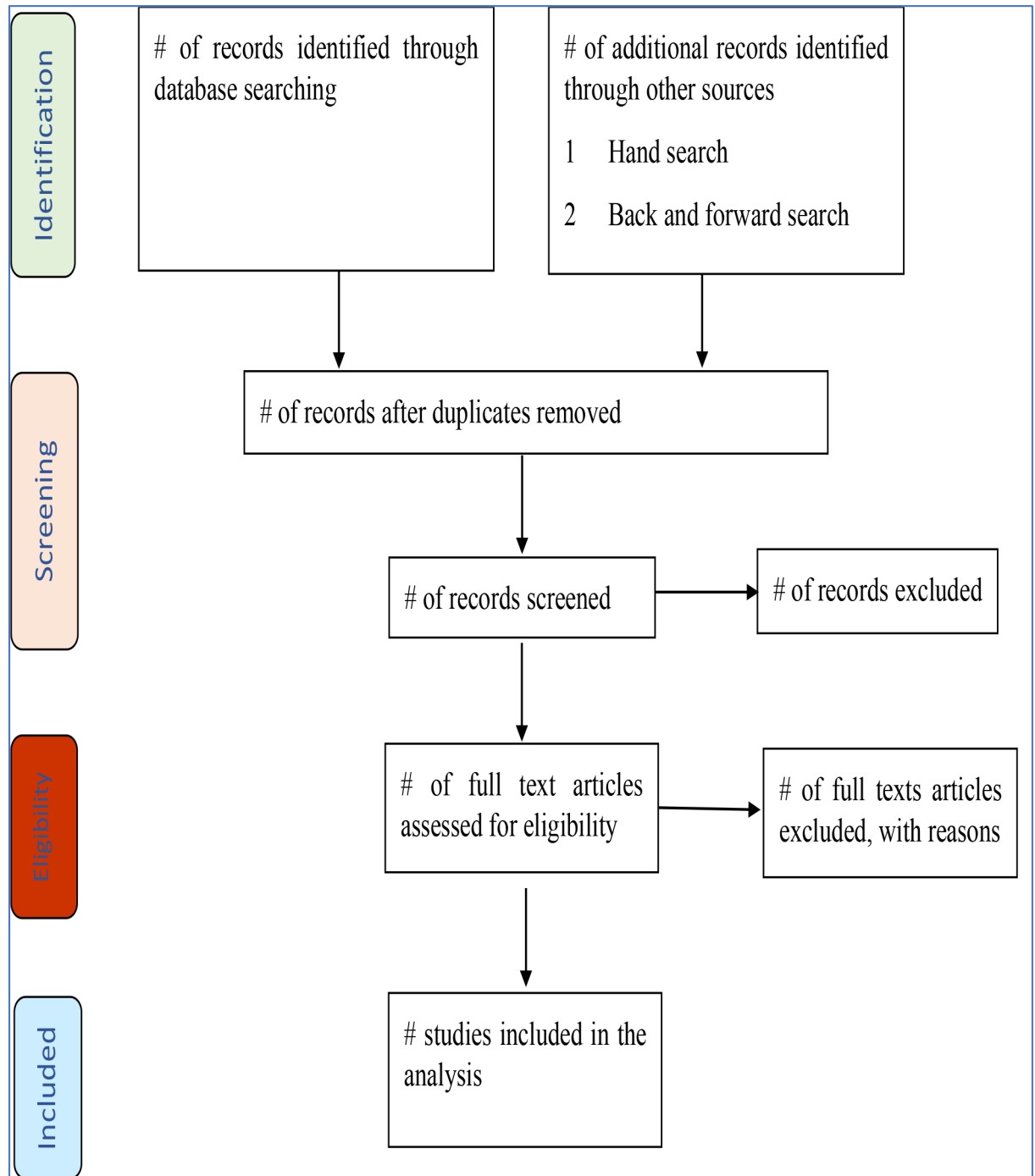

**Figure 1** Preferred Reporting Items for Systematic Reviews and Meta-analyses (PRISMA)–2020 flow diagram for the summary of the systematic review study selection process.

## Study selection and data extraction

After a comprehensive search, data will be imported to Endnote V.X7.8 (Thomson Reuters), and duplicates will be removed. Studies will be exported to Rayyan for screening by title and abstract. Two independent reviewers (TYA and EAG) will screen the title, abstract and full texts to identify eligible studies. Any inconsistencies will be resolved through consensus between the two reviewers. TYA will prepare the data extraction checklist, and data will be extracted in a Microsoft Excel (V.365) spreadsheet. The following data will be extracted from the included studies: (1) bibliographical details, name of the first author, year of publication, year of data collection, country and WHO regions; (2) demographical characteristics of participants, mean/median age, the proportion of males and the country's wealth status; (3) study characteristics, study design; sample size; type of DR-TB; comorbidities like HIV and diabetes mellitus; the total number of people examined for DR-TB by Gene Xpert, line probe assay (LPA) and/or culture; the timing of developing DR-TB, frequency of contact and location of contact (household, workplace, childcare and homeless); type of contacts (households, close and casual); and proportions of MDR-TB and XDR-TB. For a study done in multiple countries, the data from each country will be reported independently if available. The study screening and selection process are summarised in figure 1.

## Quality assessment

The Newcastle-Ottawa Quality Assessment Scale will be used to assess the quality of retrospective and prospective cohort studies.[23] The quality of cross-sectional studies will be assessed using the modified version of the Newcastle-Ottawa Quality Assessment Scale.[24] The score will classify studies into low-quality (a score between 1 and 4), moderate-quality (a score between 5 and 7) and high-quality studies (a score between 8 and 9). The quality of the included studies will be done by the two reviewers (TYA and EAG). Disagreements will be resolved by the consensus between the two reviewers.

## Data synthesis and analysis

We are interested in estimating the burden of DR-TB reported as incidence or prevalence at the global level. Stata V.17 software will be used to conduct the analysis. For incidence studies, the incidence rate will be calculated as the number of incident cases per year divided by the population at risk. Similarly, for the prevalence study, the prevalence will be calculated as the number of prevalent cases divided by the total population and expressed as a proportion. A forest plot will be generated to show individual and pooled prevalence of DR-TB cases among DR-TB contacts, 95% CI, name of the first author, publication years and study weights. A random-effects meta-analysis will be used to report the pooled estimates. The presence of heterogeneity among the included studies will be evaluated using the $I^2$ statistics and a 95% CI. An $I^2$ value close to zero indicates no observed heterogeneity, and a larger value of $I^2$ shows an increased level of heterogeneity. Heterogeneity will be considered low, moderate and high when the values are below 25%, between 25% and 75% and above 75%, respectively.[25] To identify the source of heterogeneity, subgroup analysis will be carried out by study characteristics. Moreover, meta-regression will be conducted to assure the existing source of heterogeneity. Publication bias will be assessed visually using funnel plots and statistically using Egger's regression test. A trim and fill analysis will be conducted as an adjustment if there is any publication bias.[26] A sensitivity analysis will be done by trimming low-quality studies.

## Implication of the review

DR-TB contact investigation is a top priority in DR-TB infection control, being critical for locating the source of infections as patients with smear-positive DR-TB are highly contagious. Identification of cases through contact investigation can lead to timely treatment and preventative measures to be undertaken, thereby minimising the risk of disease transmission and further reducing the burden of DR-TB in the general population. Early diagnosis and detection of DR-TB will improve treatment outcomes and reduce adverse drug reactions and complications. It will also reduce the cost for the patients and households. Overall, the study will help to achieve the three END-TB targets of 2035 (no catastrophic cost, 90% reduction in mortality and 95% reduction in patients suffering from TB) through early diagnosis and treatment.

**Acknowledgements** We would like to acknowledge Curtin University for financial funding.

**Contributors** TYA designed the study and wrote the initial draft of the manuscript. ACAC, EAG, HFW, FWS and KAA critically reviewed the final manuscript. All authors approved the final manuscript for submission.

**Funding** This work is supported by the Australian National Health and Medical Research Council (NHMRC) through an Emerging Leadership Investigator Grant APP1196549. KAA is a senior researcher at Curtin University who received the fund. TYA is also supported by Curtin University Higher Degree Research (HDR) Scholarship and acknowledges Curtin University for providing support. However, the funders had no role in the design, analysis and interpretations of findings.

**Competing interests** None declared.

**Patient and public involvement** Patients and/or the public were not involved in the design, or conduct, or reporting, or dissemination plans of this research.

**Patient consent for publication** Not applicable.

**Provenance and peer review** Not commissioned; externally peer reviewed.

**ORCID iDs**
Temesgen Yihunie Akalu http://orcid.org/0000-0002-0340-0443
Eyob Alemayehu Gebreyohannes http://orcid.org/0000-0002-0075-4553

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
