## [Reviewer comments · BMJ Open]

ARTICLE DETAILS

TITLE (PROVISIONAL)	Burden of drug-resistant tuberculosis among contacts of index cases: a protocol for a systematic review
AUTHORS	Akalu, Temesgen; Clements, Archie C. A.; Gebreyohannes, Eyob Alemayehu; Wolde, Haileab Fekadu; Shiferaw, Fasil Wagnaw; Alene, Kefyalew

VERSION 1 – REVIEW

REVIEWER	Dhungana, Hom Nath University of Technology Sydney
REVIEW RETURNED	16-Jul-2023

GENERAL COMMENTS	The proposed protocol for a systematic review seems promising to unfold the prevalence and incidence of active and latent DR-TB among contacts of DR-TB patients. Moreover, the systematic review can identify a gap in knowledge about the burden of active and latent DR-TB among contacts of DR-TB patients. In very few places, some minor grammar mistakes were found. For example, in line 201, 0 is used instead of the letter o. Overall the protocol is very informative and novel for publication.
--

REVIEWER	Tiberi, Simon Barts Health NHS Trust, Blizard Institute, Queen Mary University of London, Department of Infection
REVIEW RETURNED	22-Jul-2023

GENERAL COMMENTS	Thank you for submitting your systematic review protocol to BMJOpen the study is interesting and it's an emerging hot topic. The methodology proposed is sound. The study will be challenged by the lack of currently high quality published data that you would need to perform your review.
---

REVIEWER	Fisher, Dina University of Calgary
REVIEW RETURNED	24-Jul-2023

GENERAL COMMENTS	1. Important research question.2. Need to discuss the change in the WHO XDR/pre-XDR definitions (2021) and how those will be addressed in your planned systematic review. Will you be only reviewing articles before this definition change, reviewing articles after this definition change, or including articles with both definitions ?3. Current MDRTB treatment with BPAL/BPALM is 6-9 months so should discuss the changing treatment landscape.4. It was unclear what treatment is being referred to in the last Review Questions: Levels of adherence, treatment outcomes
---

	among contacts of DRTB cases. Are you referring to treatment of active disease identified through contact tracing and comparing it to individuals who were diagnosed not through contact tracing ? Are you referring to those identified with latent tuberculosis and provided preventive treatment ?
--	---

VERSION 1 – AUTHOR RESPONSE

Reviewer: 1

The proposed protocol for a systematic review seems promising to unfold the prevalence and incidence of active and latent DR-TB among contacts of DR-TB patients. Moreover, the systematic review can identify a gap in knowledge about the burden of active and latent DR-TB among contacts of DR-TB patients.

In very few places, some minor grammar mistakes were found. For example, in line 201, 0 is used instead of the letter o.

Overall the protocol is very informative and novel for publication. We thank the reviewer for the important comments. The grammatical errors have now been corrected in the revised version of the manuscript and the changes have been highlighted in the track change documents.

Reviewer: 2

Thank you for submitting your systematic review protocol to BMJ Open the study is interesting and it's an emerging hot topic. The methodology proposed is sound. The study will be challenged by the lack of currently high-quality published data that you would need to perform your review. We agree that the topic is new as DR-TB is an emerging public health problem. We have checked the availability of adequate studies through a preliminary search and the results showed that there are an adequate number of published studies that can address our systematic review objectives for active DR-TB. However, as the reviewer suggested, there were an inadequate number of studies from our preliminary search for latent DR-TB and we have now removed this objective in the revised version of the manuscript.

Reviewer: 3

1. Important research question.

2. Need to discuss the change in the WHO XDR/pre-XDR definitions (2021) and how those will be addressed in your planned systematic review. Will you be only reviewing articles before this definition change, reviewing articles after this definition change, or including articles with both definitions ? To address this concern, we will extract important variables from each eligible study, including the year of data collection and publication as well as drug-resistant patterns. Subgroup analysis will then be conducted based on these variables to include both XDR-TB and pre-XDR-TB.

Hence, the review will not be restricted by time and will incorporate all articles published before and after the guideline change. Instead, we will perform sub-group analysis by year of enrolment (before and after guideline change) to evaluate how the change in guideline could affect the result.

3. Current MDRTB treatment with BPAL/BPALM is 6-9 months so should discuss the changing treatment landscape. As we are investigating the prevalence and incidence of DR-TB among contacts of DR-TB patients, the duration of MDR-TB treatment will not influence the outcome of our study. The treatment of DR-TB is undergoing a long period of dynamic improvements in the management of TB. Its treatment duration was improved from long-term regimen (18-24 months including 8 months injectable drugs) to short-term (6-9 months). Similarly, there is also changing in the diagnosis landscape of DR-TB. The landscape for detection of DR-TB resistance was from sophisticated culture-based tests to rapidly diagnostic tests such as GeneXpert MTB-RIF Assay and Line probe assays. These minimizes unnecessary delay and minimizes the spread of DR-TB among its contacts by minimizing the duration of infectiousness. Hence, there is a changing landscape in DR-TB diagnosis and treatment that could significantly affect the active DR-TB and latent TB infection burden. Moreover, it could significantly improve patient adherence, enhance favourable treatment

outcomes, and minimize adverse drug reactions.

4. It was unclear what treatment is being referred to in the last Review Questions: Levels of adherence, treatment outcomes among contacts of DRTB cases. Are you referring to treatment of active disease identified through contact tracing and comparing it to individuals who were diagnosed not through contact tracing? Are you referring to those identified with latent tuberculosis and provided preventive treatment? We have now removed these in the revised version of the manuscript.

VERSION 2 – REVIEW

REVIEWER	Fisher, Dina University of Calgary
REVIEW RETURNED	05-Nov-2023
GENERAL COMMENTS	Concerns identified in previous review have been appropriately addressed.